# Examining Public Messaging on Influenza Vaccine over Social Media: Unsupervised Deep Learning of 235,261 Twitter Posts from 2017 to 2023

**DOI:** 10.3390/vaccines11101518

**Published:** 2023-09-24

**Authors:** Qin Xiang Ng, Clara Xinyi Ng, Clarence Ong, Dawn Yi Xin Lee, Tau Ming Liew

**Affiliations:** 1Health Services Research Unit, Singapore General Hospital, Singapore 169608, Singapore; 2Saw Swee Hock School of Public Health, National University of Singapore, Singapore 117549, Singapore; 3NUS Yong Loo Lin School of Medicine, Singapore 117597, Singapore; 4School of Medicine, Dentistry and Nursing, University of Glasgow, Glasgow G12 8QQ, UK; 5Department of Psychiatry, Singapore General Hospital, Singapore 169608, Singapore; 6SingHealth Duke-NUS Medicine Academic Clinical Programme, Duke-NUS Medical School, Singapore 169857, Singapore; 7Health Services and Systems Research, Duke-NUS Medical School, Singapore 169857, Singapore

**Keywords:** flu vaccine, influenza, public messaging, social media, Twitter, machine learning, topic modelling

## Abstract

Although influenza vaccines are safe and efficacious, vaccination rates have remained low globally. Today, with the advent of new media, many individuals turn to social media for personal health questions and information. However, misinformation may be rife, and health communications may be suboptimal. This study, therefore, aimed to investigate the public messaging related to influenza vaccines by organizations over Twitter, which may have a far-reaching influence. The theoretical framework of the COM-B (capacity, opportunity, and motivation component of behavior) model was used to interpret the findings to aid the design of messaging strategies. Employing search terms such as “flu jab”, “flu vaccine”, “influenza vaccine”, and ‘“ influenza jab”, tweets posted in English and by organizations from 1 January 2017 to 1 March 2023 were extracted and analyzed. Using topic modeling, a total of 235,261 tweets by organizations over Twitter were grouped into four main topics: publicizing campaigns to encourage influenza vaccination, public education on the safety of influenza vaccine during pregnancy, public education on the appropriate age to receive influenza vaccine, and public education on the importance of influenza vaccine during pregnancy. Although there were no glaring pieces of misinformation or misconceptions, the current public messaging covered a rather limited scope. Further information could be provided about influenza and the benefits of vaccination (capability), promoting community, pharmacist-led influenza vaccination, and other avenues (opportunity), and providing greater incentivization and support for vaccination (motivation).

## 1. Introduction

The global uptake of influenza vaccines varies across different regions of the world. Despite the wealth of evidence supporting the safety and efficacy of influenza vaccination [1], vaccination rates have remained persistently low; influenza vaccine coverage was as low as 1.5% to 2.2% in China [2] and 14% in Singapore [3], while it failed to cross the 50% threshold in other developed countries like Germany [4] and the United States [5]. Historically, many countries struggle with low influenza vaccination coverage and fall short of the target immunization coverage. This deficiency is further exacerbated by the Coronavirus Disease 2019 (COVID-19) epidemic, which, despite the availability of novel and effective vaccines, has seen unprecedented levels of vaccine hesitancy or refusal [6]. Unfortunately, the aversion towards vaccination within a single disease may indicate potential negative spill-over effects, as witnessed by the decline in influenza and other childhood immunizations [7,8].

Given that flu causes up to 650,000 deaths annually, influenza poses a grave threat to public health. In view of this problem, flu vaccination plays a crucial role in preventing millions of illnesses and flu-related doctor visits each year and can reduce the incidence of influenza significantly [9,10]. Notably, there is also a particular urgency as the CDC’s monitoring systems for tracking in-season flu vaccination rates have identified worrisome patterns. The current season has seen lower influenza vaccination coverage for most demographic groups compared to the previous season, including segments at elevated risk of severe flu consequences or complications, such as pregnant individuals and other susceptible groups [9].

Today, with the advent of new media, social media platforms have emerged as popular sources of personal health information and advice for individuals. With more than 200 million active daily users [11], Twitter stands out as a powerful tool for the dissemination of information. However, misinformation, particularly on public health issues such as vaccination, has been found to be prevalent on these platforms [12], with the COVID-19 pandemic further fueling the proliferation of misinformation [13]. Given the contribution of misinformation on social media to vaccine hesitancy [14], it is crucial to implement countermeasures to tackle this problem. Current health communications related to the influenza vaccine may also be suboptimal in promoting uptake.

Yet, despite the considerable potential impact of such messaging, there is a dearth of research that specifically investigates the public communication pertaining to influenza vaccines as conveyed by organizations on social media. Organizations with a presence on Twitter may hold profound influence in shaping the topic of public agenda [15] and setting the agenda with regard to vaccination programs [16]. As entities within respective communities, organizations actively leverage social learning [17] and social identities [18] to establish norms while influencing attitudes and behaviors. Studying how online public messages are crafted and delivered can offer valuable insights into the health communication practices of these organizations and how they could be improved. This knowledge can then be applied to other public health initiatives, such as promoting other vaccines and raising awareness about disease prevention.

In this study, we, therefore, investigate the discourse surrounding influenza vaccination by employing an unsupervised deep-learning analysis of publicly available Twitter posts from 2017 to 2023. In particular, we chose to target tweets from organizations, which are generally regarded by the public as more authoritative sources of information than personal, subjective viewpoints. Given the likely large corpus of tweets for analysis, unsupervised machine-learning techniques were applied to facilitate the process.

We hypothesize that the current public messaging predominantly emphasizes the importance of influenza vaccination, its benefits, and accessibility, with the aim of encouraging individuals, especially those at higher risk of complications from the flu, to be vaccinated. Through the analysis of public messaging, we seek to glean insights into its strengths and areas for improvement and to uncover potential new avenues and opportunities to bolster influenza vaccine uptake.

## 2. Methodology

The approach used in this study’s methodology was derived from prior infodemiology research, which underscored the application of machine learning methodologies for analyzing unstructured free-text content on Twitter [19,20,21] so as to investigate public sentiments and comprehend public conversations related to a specific subject or topic. Unsupervised deep learning allows for the analysis of large datasets without predefined labels or categories. This approach is particularly useful when exploring complex, multifaceted topics like public sentiments and reactions to vaccine messaging where pre-defined categories might limit the depth and breadth of insights.

Opting for Twitter as the selected social media platform and employing the search terms “flu jab”, “#flujab”, “flu vaccine”, “#fluvaccine”, “influenza vaccine”, “#influenzavaccine”, “influenza jab”, and “#influenzajab”, we collected English-language tweets from 1 January 2017 to 1 March 2023. As vaccine hesitancy and acceptance are dynamic rather than static concepts and fluctuate over time [22], the timeframe from 2017 to 2023 was chosen because it is recent and would also allow us to study the concurrent timeline of the COVID-19 pandemic and related vaccine drives. This period captures the years just before the pandemic and during its major waves, enabling an analysis of the potential influence of COVID-19 on influenza vaccine messaging. The data used in this study was also extracted prior to the new changes to Twitter’s algorithm and its rebranding to X Corp in April this year. Extraction took place via Twitter’s Application Programming Interface (API) using an academic developer account, allowing the download of up to 10 million tweets per month without sampling. Retweets and duplicate tweets were excluded from the analysis. The study covered a global scope, with no restriction on tweet origin country.

Focusing on influenza vaccine-related public messaging, only organizational tweets were considered and were identified using BERT Named Entity Recognition (NER) [23]. BERT NER has been trained using a pre-training and fine-tuning approach and uses a sequence labeling approach to identify named entities such as organizations, individuals, and locations within unstructured free text. It is able to recognize four types of entities: location (LOC), organizations (ORG), person (PER), and miscellaneous (MISC). Subsequently, BERTopic, a topic modeling technique, employed BERT embeddings and class-based TF-IDF to generate interpretable topics concerning public messaging on influenza vaccines. These topics highlighted the keywords in the tweets encompassed.

BERTopic, a topic modeling technique that leverages state-of-the-art Bidirectional Encoder Representations from Transformers (BERT) embeddings and class-based term frequency–inverse document frequency (TF-IDF) [24], was then applied to generate interpretable topics around the public messaging on influenza vaccines and highlight the keywords in the topic descriptions.

We also calculated the mean public attention scores for each topic based on the sum of the retweet count, reply count, like count, and quote count, and expressed as a numerical score, as adapted from previous public opinion research [21].

No human participants were directly involved in this study. Ethical clearance was obtained from the SingHealth Centralised Institutional Review Board of Singapore (reference number: 2021/2717). Data collection adhered to Twitter’s terms of use and standard academic practices when using social media for research.

## 3. Results

### 3.1. Retrieval of Relevant Tweets

A dataset comprising 1,153,059 tweets was compiled, of which 983,299 unique tweets were composed in English. Following the exclusion of tweets authored by individual users, a subset of 235,261 tweets originating from organizational accounts on the Twitter platform was analyzed (Figure 1).

Figure 2 displays a map indicating the macro-areas from which the tweets originated. As anticipated, most of the tweets originated in North America (*n* = 72,948, 31.0%) and Europe (*n* = 69,410, 29.5%), aligning with the overall demographic of English-speaking Twitter users.

### 3.2. Topic Modeling

The public messaging related to flu vaccines appeared to center around four topics based on topic modeling: (1) publicizing campaigns to encourage influenza vaccination, (2) public education on the safety of influenza vaccines during pregnancy, (3) public education on the appropriate age at which to receive the influenza vaccine, and (4) public education of the importance of the influenza vaccine during pregnancy. Topic 1 constituted the bulk of the tweets (*n* = 218,093, 92.7%), which contained public advertisements about vaccination campaigns.

Topic 1 publicizes campaigns to receive the influenza vaccine and also highlights individuals, including staff members, CEOs, healthcare assistants, and others actively participating in the flu vaccination campaign. Key figures such as CEOs, medical directors, and chief nurses are shown leading by example, being vaccinated, and encouraging others to do the same. The sense of unity and teamwork is also promoted by showcasing various individuals receiving their influenza jabs. Hashtags such as #flujab, #flu, #FluVaccine, #FluFighters, #FluSeason, and #FluFighter are used to emphasize the vaccination campaign. The tweets encourage people to protect themselves and their patients by being vaccinated, highlighting that even healthcare professionals are taking part.

Meanwhile, Topic 2 underscores the safety of receiving the influenza vaccine during pregnancy for both the pregnant parent and the baby. Several tweets cite statistics and research findings related to influenza vaccine usage during pregnancy and reference the good safety profile and lack of negative health outcomes associated with influenza vaccination during pregnancy. Related to this emphasis, Topic 4 highlights the efficacy of the vaccine in preventing illness in pregnant women and protecting newborns. The potential severity of illness for both the mother and the baby is emphasized as a reason to prioritize vaccination. The tweets describe potential complications that may arise from not getting vaccinated, such as premature labor, low birth weight, and increased risk of flu-related complications.

Last, Topic 3 highlights the target audience and eligibility for the influenza vaccine, which is recommended for everyone aged six months and older. Specific vulnerable groups, such as children lacking immunity to the flu and individuals over 65, are emphasized as being particularly susceptible to the flu.

We also computed the public attention scores for each topic, taken as the sum of the retweet count, reply count, like count, and quote count. As seen in Table 1, Topic 3 (Public education on the appropriate age to receive influenza vaccine) rates the highest mean public attention score, indicating that tweets on this topic are more likely to be retweeted, replied to, liked, or quoted by the public. This score may possibly indicate that the information aligns well with user interests, leading to widespread sharing, or it might highlight an area of common uncertainty among the general populace. A sample of the top ten tweets that have received much public attention for each topic is detailed in the Appendix A.

## 4. Discussion

We utilized unsupervised deep learning to examine a substantial amount of unstructured text data from social media. The aim was to gain insight into the prevalent public discourse concerning influenza vaccination. From this analysis, four main themes of public messaging were discerned. Importantly, our investigation indicated that tweets from various organizations on Twitter did not prominently showcase misinformation or misconceptions. This observation is noteworthy, given the substantial occurrence of health-related misinformation on Twitter, especially concerning significant public health matters [12]. This finding partially corresponds to earlier research, which found that while the prevalence of low credibility information, particularly regarding the COVID-19 pandemic, was on average 32% in 2020, with a large volume of tweets originating from verified accounts [13], the majority of the misinformation originated from individuals, including ordinary users and even prominent public figures, who demonstrated significantly greater social media engagement than others [25]. The common themes of misinformation identified in prior research address conspiracy theories, safety, and efficacy concerns [26]. Such misinformation commonly misrepresented vaccine studies by drawing false conclusions, deceptive use of sources, selective data description, and unsupported claims [27]. This discrepancy may be attributed to the more stringent scrutiny of social media posts from organizations by public health agencies or governmental bodies. Nevertheless, the worrisome fact remains that low-credibility sources with limited trustworthiness could achieve similar reshare volumes to mainstream sources and, in some cases, even surpass authoritative sources like the CDC [28]. This possibility underscores the urgent need to strengthen efforts in promoting accurate vaccination messages by reputable organizations through enhanced outreach strategies.

The findings from this study support our initial hypothesis that the current public messaging predominantly emphasizes the importance of influenza vaccination, its benefits, and accessibility, with the aim of encouraging individuals, especially those at higher risk of complications from the flu, to get vaccinated. Nonetheless, there are missed opportunities here, and the present findings suggest practical implications for public health practitioners. Mapping these findings into the theoretical framework of the COM-B (capacity, opportunity, and motivation component of behavior) model [29] can provide further insight into the strengths and limitations of current public messaging strategies related to influenza vaccines. The COM-B model is a theoretical framework for behavior change that suggests that for an individual to engage in a desired behavior, they must have the capability (C), opportunity (O), and motivation (M) to do so [29]. Capability refers to both the psychological and physical ability to perform the behavior; opportunity refers to the presence of external factors that allow for the behavior to occur; and motivation refers to the internal drives and desires to change. While human behavior is complex and influenced by various factors, the COM-B model identifies specific targets for intervention to guide efforts in increasing influenza vaccine uptake.

Overall, the identified topics in this study align well with the different elements of the COM-B model of behavior change [29]. However, there remain areas for improvement in current public messaging strategies. In the context of influenza vaccine uptake, the construct of “capability” refers to an individual’s physical and psychological ability to access and receive the influenza vaccine. Capability encompasses factors such as understanding the benefits of vaccination, having knowledge about where to get vaccinated, and having the time and resources to receive the vaccine. It is worth noting that although Topic 4 and some tweets in Topic 3 provided public education by addressing the timing and importance of influenza vaccination during pregnancy, these topics constituted only a small proportion of the overall tweets. Moreover, there is a significant lack of emphasis on the broader benefits of vaccination, such as herd immunity, in general public messaging on social media. Receiving the influenza vaccine not only protects individuals but also contributes to the health and well-being of the entire community. Individuals within communities are moral actors [30], and the saliency of a collective identity has been shown to encourage individual cooperation and contribution [31]. In fact, collective frames that target altruism and a sense of community were found to have a positive relationship with stated vaccination intentions [32].

Likewise, the construct of opportunity relates to external elements that influence an individual’s capacity to receive the vaccine, including its availability, accessibility of vaccination sites, and associated costs. However, the public messaging on Twitter seems to be primarily focused on visiting primary care providers to obtain the influenza vaccine. This narrow focus represents missed opportunities, particularly in countries with structural impediments, distant geographic locations, and a large population, where primary care providers may be scarce and inconveniently located [33]. To enhance vaccine uptake, public messaging could adopt a more proactive and diverse approach, specifically highlighting how and where members of the public can receive influenza vaccines. Including additional information, such as the location of vaccination centers, reduces the cognitive effort required for individuals to find this information themselves, making the vaccination process more accessible, tangible, and less difficult [34]. In addition to nudging the public to visit their general practitioners, where available, the public messaging can also promote the accessibility of influenza vaccines at community pharmacies. Previous studies have shown that community pharmacists provide a convenient and accessible option for seasonal influenza vaccination, and such an approach has been positively received by patients [29,35].

The concept of motivation relates to an individual’s intrinsic drive to receive the influenza vaccine. It encompasses various factors, including the perceived susceptibility to influenza, the perceived severity of influenza, and the perceived benefits of the influenza vaccination. Additionally, prevailing social norms, attitudes, and beliefs about vaccination contribute to shaping an individual’s motivation. In our analysis, Topic 1 featured examples of individuals who had received the influenza vaccine and encouraged others to follow suit, reflecting a form of role modeling and imitation consistent with social learning theory [36]. On the other hand, Topic 2 addressed public concerns, specifically on the safety of influenza vaccines during pregnancy. However, there appears to be relatively less focus on the safety of the influenza vaccine for other population groups. Notably, vulnerable populations such as those with multimorbidity and the elderly were conspicuously missing from the conversation. It is crucial to highlight the benefits of influenza vaccination even for healthy working adults, emphasizing the potential reduction in influenza-like illnesses, physician visits, and work absenteeism [1]. Previous studies have demonstrated that presenting factual, evidence-based messages that include cost-benefit comparisons in a transparent manner, especially when addressing concerns of target populations, such as vaccine safety, can have a positive impact on public beliefs and intentions [37,38]. By expanding the scope of safety discussions and promoting vaccination benefits across various population segments, public messaging can potentially be effective in motivating a broader range of individuals to consider influenza vaccination.

On the whole, upon examining the clustering of tweets, it is evident that the dominant topic identified in our analysis centers around “campaigns.” The prominence of this topic could be attributed in part to the COVID-19 pandemic, which spanned a significant portion of the study’s duration. Amidst the pandemic, there was a widespread push for promoting the uptake of COVID-19 vaccines, followed closely by advocacy for seasonal flu shots in many countries. For the 2021-2022 influenza season, the World Health Organization (WHO) and numerous European countries promoted the safe and efficacious coadministration of influenza and COVID-19 vaccines [39]. In the realm of public health issues, traditional mass media campaigns have been shown to be effective in driving change, particularly when combined with the availability and access to key services [40]. Social media platforms have emerged as powerful tools for promoting such campaigns, with health behavior change interventions on these platforms generally showing positive outcomes [41]. However, previous research highlights that while health agencies utilize Twitter for information dissemination, only a small fraction of tweets garner significant engagement, as measured by retweet count [42]. To maximize engagement, specific strategies can be employed to craft tweets, such as incorporating hashtags, user mentions, and uniform resource locators (URLs) [42]. Moreover, involving target communities in the development of messaging strategies can enhance engagement [43]. In addition to disseminating information, monitoring public opinion and sentiment on social media is crucial. This monitoring enables the adaptation and modification of strategies to address evolving challenges, thereby enhancing flu campaigns and increasing vaccine uptake. However, research gaps still exist concerning the actual impact of such social media campaigns on health interventions [44]. Future studies should evaluate the efficacy of these campaigns in influencing individual behaviors and increasing vaccine uptake.

The findings of the present study should be interpreted in light of some limitations. First, we exclusively focused on analyzing Twitter to the exclusion of other social media platforms. The user composition on Twitter predominantly comprises individuals and organizations from North America, Europe, and the United Kingdom; i.e., the user demographic on Twitter does not represent the global populace. This potential sampling bias may restrict the applicability of our findings to more diverse populations. Moreover, there are also vaccine users who do not use social media. Related to this fact, only English-language tweets were analyzed in this study because English is one of the most widely used languages on Twitter, and focusing on a single language ensures consistency in language processing techniques, thereby reducing the complexity introduced by multilingual data analysis. However, it is important to note that this choice may introduce a bias towards English-speaking regions and their specific cultural, political, and health contexts [45]. Second, the tweets used in the present dataset were filtered based on specific keyword searches, potentially leading to some selection biases. The search criteria might predominantly spotlight tweets from mainstream organizations endorsing vaccination. As an illustration, tweets employing hashtags like #BigPharma and #StopMandatoryVaccines could disseminate misleading information or demonstrate “anti-vax” stances on flu shots without directly mentioning “flu” or “influenza.” Such tweets might be overlooked in our search parameters. This factor could mean our analysis might not capture these nuanced discussions and alternate messages. Third, as the study utilizes an unlabeled dataset, unsupervised learning is inherently more difficult than supervised learning, and the results of the topic modeling might be less accurate than otherwise, as the output is not known. Fourth, Twitter’s imposed character limit (previously 140, now 280) restricts the amount of information that can be shared in a single tweet. For public messaging, this limit has inherent constraints and, in our analyses, can lead to oversimplification or loss of context, creating challenges for capturing and conveying certain nuanced information regarding influenza vaccines and vaccine hesitancy. Human language is not always explicit, and tweets are limited. Fifth, as the data used in this study was extracted prior to the new changes to Twitter’s algorithm and its rebranding to X Corp in April this year, it is unclear how these changes might affect marketers and the adoption and use of the platform in the future.

## 5. Conclusions

In summary, this study delved into the discourse surrounding influenza vaccines, specifically focusing on Twitter posts from organizations. The study offers a structured and comprehensive analysis of how influenza vaccination is discussed on a major social media platform during a significant period in global health, providing valuable data for future health communication strategies. While the analysis did not reveal any significant clusters of tweets containing misinformation or misunderstandings, the prevailing public messaging exhibited a narrow scope. Future messaging should encompass broader aspects, including disseminating comprehensive knowledge about influenza and the wide-ranging advantages of vaccination (capability), advocating for diverse means of vaccine access (opportunity), and bolstering motivation through greater incentivization and social support for vaccination (motivation).

## Figures and Tables

**Figure 1 vaccines-11-01518-f001:**
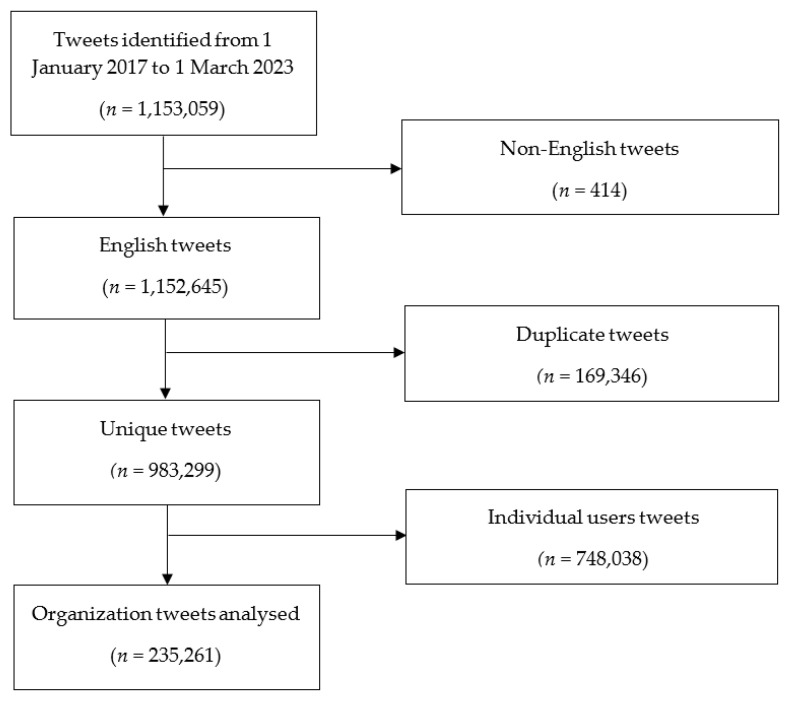
Flowchart showing the selection process for relevant tweets.

**Figure 2 vaccines-11-01518-f002:**
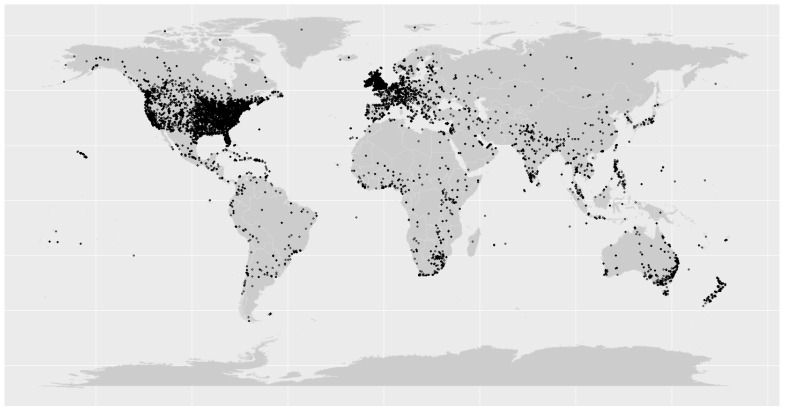
The geographical positions of the tweets encompassed within this study are visually represented on the world map (with each individual tweet marked by a black dot).

**Table 1 vaccines-11-01518-t001:** Topic labels, keywords, and sample tweets pertaining to the public messaging for influenza vaccines, based on BERTopic.

Topic Label *(keywords)*	Sample Tweets	Number of Tweets, *n* (%)	Public Attention Score, Mean (SD) ^1^
Topic 1: Publicizing campaigns to encourage influenza vaccination (*flujab, im, swine, swine flu, like flu vaccine, deaths, 50, swine flu vaccine, universal, year flu*)	“Our staff #FluJab campaign has begun! Take a look at our Acting Chief Executive: John Holden, Chief Nurse: Karen Dawber; Chairman: Max Mclean getting their vaccine. For children its a simple nasal spray. Have you had yours? #FluFighter #FluSeason #Bradford #BeInfluential” “More than 70 staff had their #flujab yesterday, including Melanie Walker and Chair Julie Dent. The team have been busy at Our Journey events so far! #DPTOurJourney #FluFighters #flu #FluSeason #FluFighter #FluVaccine #NHS”	218,093 (92.7)	5.3 (104.2)
Topic 2: Public education on the safety of influenza vaccine during pregnancy (*egg, eggs, miscarriage, vaccine pregnancy, allergic, allergy, flu vaccine pregnancy, egg allergy, vaccine miscarriage, women flu vaccine*)	“#influenza Influenza vaccine in pregnancy is not associated with stillbirth. Of 795 stillbirths, 43.1% of women received the flu vaccine in pregnancy; and of 3180 live births, 44.3% of women received the flu vaccine in pregnancy.” “Allergy to egg is no longer a contraindication for getting the flu vaccine. Extensive testing updated the recommendation.”	1780 (0.8)	5.3 (16.9)
Topic 3: Public education on the appropriate age to receive influenza vaccine (*months older, older flu, older flu vaccine, months older flu, age older, months age older, recommends months, cdc recommends, age months, recommends months older*)	“Its Flu Season. Everyone 6 months and older should get an annual flu vaccine. One Community Health is here to help you get your flu vaccine come see us. #onecommunityhealth #healthytogether” “Because one’s immune response from previous vaccination wanes over time and updated formulation of the influenza vaccine becomes available, people aged 6 months and older should receive the influenza vaccine each year. #aware”	2031 (0.9)	9.1 (70.0)
Topic 4: Public education on the importance of influenza vaccine during pregnancy (*midwife, free flu jab, baby flu, pharmacist midwife, gp pharmacist midwife, pregnant flu, gp midwife, youre pregnant, baby, stage pregnancy*)	“This year it is even more important for mums-to-be to have their annual #FluVaccine. If you are pregnant and catch flu and COVID-19 at the same time, it could make you and your baby seriously ill. #FreeBecauseYouNeedIt #Flu” “Getting flu while pregnant, may cause premature labour, or it may result in a baby having a low birth weight. Ask your GP for a #flu jab. #StayWell #HelpUsHelpYou”	1668 (0.7)	4.0 (11.5)

^1^ Public attention score for each tweet was calculated as the sum of the retweet count, reply count, like count, and quote count.

## Data Availability

The datasets generated during and/or analyzed during the current study are available from the corresponding author on reasonable request.

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
