# Peer review of "Examining Public Messaging on Influenza Vaccine over Social Media: Unsupervised Deep Learning of 235,261 Twitter Posts from 2017 to 2023"

_vaccines, 2023, doi:10.3390/vaccines11101518_

Round 1

Reviewer 1 Report

The authors have conducted a study related to public messages using deep supervised learning. The content is quite interesting and nicely written manuscript. 

The manuscript entitled "Examining public messaging on influenza vaccine over social 2 media: unsupervised deep learning of 235,261 Twitter posts 3 from 2017 to 2023". The content is quite interesting.

The abstract, introduction, and methodology sections are well written still some lacunae have been observed. 

The authors have chosen 2017 to 2023 but the reason is not clearly mentioned in the entire manuscript. The authors should incorporate the reasons behind choosing the year frame.

The authors should explain the novelty of the content that is missing. Authors should incorporate this in their revised manuscript.

Authors should explain why the unsupervised deep learning method they have adapted for this study is not clearly stated in the manuscript.

The authors should incorporate some more limitations of the study. Already they have provided but not sufficient.

The authors should rewrite the conclusion section 

The authors should address the comments and incorporate them in the revised manuscript. Although the study is interesting, keeping in mind the standard of the journal revision of the manuscript is of utmost importance. Furthermore, I recommend a major revision.

Author Response

Reviewer 1

Comment 1: The authors have conducted a study related to public messages using deep supervised learning. The content is quite interesting and nicely written manuscript. The manuscript entitled "Examining public messaging on influenza vaccine over social 2 media: unsupervised deep learning of 235,261 Twitter posts 3 from 2017 to 2023". The content is quite interesting. The abstract, introduction, and methodology sections are well written still some lacunae have been observed.

Reply 1: Thank you for the kind words!

Comment 2: The authors have chosen 2017 to 2023 but the reason is not clearly mentioned in the entire manuscript. The authors should incorporate the reasons behind choosing the year frame.

Reply 2: Thank you for the comment. We have now explained in the methods section that, “As vaccine hesitancy and acceptance are dynamic rather than static concepts and fluctuate over time [22], the timeframe from 2017 to 2023 was chosen because it is more recent and would also allow us to study the concurrent timeline of the COVID-19 pandemic and related vaccine drives. This period captures the years just before the pandemic and during its major waves, enabling an analysis of the potential influence of COVID-19 on influenza vaccine messaging. The data used in this study was also extracted prior to the new changes to Twitter’s algorithm and its rebranding to X Corp in April this year.”

Comment 3: The authors should explain the novelty of the content that is missing. Authors should incorporate this in their revised manuscript.

Reply 3: Thank you for the comment. We have further highlighted the novelty of our study in the conclusion section, “The study offers a structured and comprehensive analysis of how influenza vaccination is discussed on a major social media platform during a significant period in global health, providing valuable data for future health communication strategies.”

Comment 4: Authors should explain why the unsupervised deep learning method they have adapted for this study is not clearly stated in the manuscript.

Reply 4: Thank you for the comment. We have now explained in the methods section that, “Unsupervised deep learning allows for the analysis of large datasets without predefined labels or categories, and this approach is particularly useful when exploring complex, multifaceted topics like public sentiments and reactions to vaccine messaging where pre-defined categories might limit the depth and breadth of insights.

Comment 5: The authors should incorporate some more limitations of the study. Already they have provided but not sufficient.

Reply 5: Thank you for the comment. We have added additional limitations to the discussion section as suggested. “First, we exclusively focused on analysing Twitter to the exclusion of other social media platforms. The user composition on Twitter predominantly comprises individuals and organisations from North America, Europe, and the United Kingdom, i.e., the user demographic on Twitter does not represent the entire populace. This potential sampling bias may restrict the applicability of our findings to more diverse populations. Moreover, there are also vaccine users who do not use social media. Related to this, only English-language tweets were analyzed in this study because English is one of the most widely used languages on Twitter and focusing on a single language ensures consistency in language processing techniques and reduces the complexity introduced by multilingual data analysis. However, it is important to note that this choice may introduce a bias towards English-speaking regions and their specific cultural, political, and health contexts [44]. Second, the tweets used in the present dataset were filtered based on specific keyword searches, potentially leading to some selection biases. The search criteria might predominantly spotlight tweets from mainstream organisations endorsing vaccination. As an illustration, tweets employing hashtags like #BigPharma and #StopMandatoryVaccines could disseminate misleading information or demonstrate ‘anti-vax’ stances on flu shots without directly mentioning “flu” or “influenza”. Such tweets might be overlooked in our search parameters. This could mean our analysis might not capture these nuanced discussions and alternate messages. Third, as the study utilises an unlabelled dataset, unsupervised learning is inherently more difficult than supervised learning and the results of the topic modelling might be less accurate as the output is not known. Fourth, Twitter’s imposed character limit (previously 140, now 280) restricts the amount of information that can be shared in a single tweet. In terms of public messaging, this has inherent constraints, and in our analyses, can lead to oversimplification or loss of context, making it challenging to capture and convey certain nuanced information regarding influenza vaccines and vaccine hesitancy. Human language is not always explicit and tweets are limited. Fifth, as the data used in this study was extracted prior to the new changes to Twitter’s algorithm and its rebranding to X Corp in April this year, it is unclear how these may affect marketers and the adoption and use of the platform in the future.”

Comment 6: The authors should rewrite the conclusion section.

Reply 6: We have rewritten the conclusion section as suggested. The revised conclusion now reads, “In summary, this study delved into the discourse surrounding influenza vaccines, specifically focusing on posts from organisations on Twitter. The study offers a structured and comprehensive analysis of how influenza vaccination is discussed on a major social media platform during a significant period in global health, providing valuable data for future health communication strategies. While the analysis did not reveal any significant clusters of tweets containing misinformation or misunderstandings, the prevailing public messaging exhibited a narrow scope. Future messaging should encompass broader aspects, including disseminating comprehensive knowledge about influenza and the wide-ranging advantages of vaccination (capability), advocating for diverse means of vaccine access (opportunity), and bolstering motivation through greater incentivisation and social support for vaccination (motivation).”

Reviewer 2 Report

The manuscript by Liew and co-workers present, based on Twitter data, that health organizations support influenza vaccines. Flu vaccination is an important topic, especially – I think – the balance between medically sound, scientific advice versus conspiracy theories and false information. The manuscript while references the literature on false vaccine information, it did not try to collect such data or identify where they originate from, and to what extent do they penetrate society. I think it is a missed opportunity. I did not expect anything surprising to come out of analyzing health organization, especially English-speaking ones (it is still reassuring that those organizations take their job seriously).

Is the list of tags and search terms the whole list of them? Considering that the sentence has a “employing search terms such as” phrase, it gives the feeling that there are many more search terms, but the authors only show these as examples. I kindly ask the authors to have the full list of search terms in the method section.

What search terms would the authors use if they wanted to sample conspiracy theorists, vaccine hesitant and anti-vaxers as well and not only organizations supporting vaccination?

I was unable to look up what kind of data one can get from Twitter. The body of the tweet (text) and the date are given, but do the researchers also have access to likes or retweet counts? Can retweets be estimated based o the number of times a tweet is found? Are duplicates retweets? Retweets might give an estimate on reach, and also on the spreading of information. It would be nice if we would know if these messages from the organizations are just paying lip service to pro-vaccine marketing, but ultimately reaches very few people, or are they part of a meaningful information campaign?

The discussion of limitations should also address the selection of search terms and why only English was used in the search. If the reason is that most Tweets are in English, then why not limit the search to only English-speaking countries? What does it tell us, that there are pro-vaccine, English tweets in France, Italy, etc.?

Minor comments:

Line43: space needed between States and [5]

Table 1.: I suggest using topic as the heading of the first column instead of “Table .

Author Response

Reviewer 2

Comment 1: Is the list of tags and search terms the whole list of them? Considering that the sentence has a “employing search terms such as” phrase, it gives the feeling that there are many more search terms, but the authors only show these as examples. I kindly ask the authors to have the full list of search terms in the method section.

Reply 1: Apologies for the ambiguity. Yes, the list of tags and search terms given represent the whole list of them. We have now reworded the sentence to, “Opting for Twitter as the selected social media platform, and employing the search terms ‘flu jab’, ‘#flujab’, ‘flu vaccine’, ‘#fluvaccine’, ‘influenza vaccine’, ‘#influ-enzavaccine’, ‘influenza jab’, and ‘#influenzajab’, we collected English-language tweets from January 1, 2017, to March 1, 2023.”

Comment 2: What search terms would the authors use if they wanted to sample conspiracy theorists, vaccine hesitant and anti-vaxers as well and not only organizations supporting vaccination?

Reply 2: Thank you for this interesting comment. Yes we have tried other search terms and in particular, we found that there are tweets with hashtags such as #BigPharma and #StopMandatoryVaccines, which convey strongly negative and ‘anti-vax’ attitudes towards influenza vaccination without directly referencing the word “flu” or “influenza”. While we tried to capture a range of tweets with our preset keywords, there is definitely a possibility a proportion of tweets from non-mainstream organizations may not contain these specific words. We acknowledge this limitation in our discussion section, “Second, the tweets used in the present dataset were filtered based on specific keyword searches, potentially leading to some selection biases. The search criteria might predominantly spotlight tweets from mainstream organisations endorsing vaccination. As an illustration, tweets employing hashtags like #BigPharma and #StopMandatoryVaccines could disseminate misleading information or demonstrate 'anti-vax' stances on flu shots without directly mentioning “flu” or “influenza”. Such tweets might be over-looked in our search parameters. This could mean our analysis might not capture these nuanced discussions and alternate messages.”

Comment 3: I was unable to look up what kind of data one can get from Twitter. The body of the tweet (text) and the date are given, but do the researchers also have access to likes or retweet counts? Can retweets be estimated based on the number of times a tweet is found? Are duplicates retweets? Retweets might give an estimate on reach, and also on the spreading of information. It would be nice if we would know if these messages from the organizations are just paying lip service to pro-vaccine marketing, but ultimately reaches very few people, or are they part of a meaningful information campaign?

Reply 3: Thank you for the comments and suggestions. We have now calculated the mean public attention scores for each topic, taken as the sum of the retweet count, reply count, like count and quote count, and expressed as a numerical score, as adapted from previous public opinion research. We have also added in our Supplementary Material a sample of the top 10 tweets that have received much public attention for each topic. As explained by the reviewer, if a tweet is retweeted multiple times, it may suggest that the content is resonating with users and is being widely shared.

Comment 4: The discussion of limitations should also address the selection of search terms and why only English was used in the search. If the reason is that most Tweets are in English, then why not limit the search to only English-speaking countries? What does it tell us, that there are pro-vaccine, English tweets in France, Italy, etc.?

Reply 4: Thank you for the comments. English-language tweets were primarily analyzed because English is one of the most widely used languages on Twitter and offers a broad perspective on global sentiments. Additionally, focusing on a single language ensures consistency in language processing techniques and reduces the complexity introduced by multilingual data analysis. Unfortunately, we could not reliably restrict the tweets to only certain countries as although we can extract geolocational data for users who do share it, this is not always reliable and Twitter also removed the option to share the geolocation and coordinates of tweets since 2019. We acknowledge these limitations in our discussion of study limitations.

Comment 5: Line43: space needed between States and [5]

Reply 5: Thank you for noticing this. We have added the missing space.

Comment 6: Table 1.: I suggest using topic as the heading of the first column instead of “Table.”

Reply 6: Thank you for noticing this. We have now changed ‘Table’ to ‘Topic’ as suggested.

Reviewer 3 Report

I have read and revised the manuscript entitled "  Examining public messaging on influenza vaccine over social 2 media: unsupervised deep learning of 235,261 Twitter posts 3 from 2017 to 2023 "

Without a doubt it seems to me a very novel methodology to know at the level of the general population some social debates about vaccines.

The writing is good, the text is well structured and the different sections of the manuscript well developed.

Although they treat COVID lightly, I think they could devote more emphasis to how the SARS COV 2 pandemic time has been able to influence their results, which has also occupied almost half of their study time.

 Despite the novelty of its methodology, the results are quite predictable and I do not consider them to contribute novelties to the current state of the subject. Can as authors justify that practical contribution is derived from the results of their study?

 The exclusion of tweets written in Spanish excludes almost half of the world's population and is an important limitation of their study. 

Other limitations exist with this study. Human language is not always explicit, and tweets are limited.

In addition, these data were collected using keyword searches and may have missed participants.

From the ethical point of view it is understood that all users of social networks authorize their content to be analyzed, however there are many vaccine users who do not use social networks to comment on them. There is significant selection bias

Author Response

Reviewer 3

Comment 1: I have read and revised the manuscript entitled “Examining public messaging on influenza vaccine over social 2 media: unsupervised deep learning of 235,261 Twitter posts 3 from 2017 to 2023”. Without a doubt it seems to me a very novel methodology to know at the level of the general population some social debates about vaccines. The writing is good, the text is well structured and the different sections of the manuscript well developed. Although they treat COVID lightly, I think they could devote more emphasis to how the SARS COV 2 pandemic time has been able to influence their results, which has also occupied almost half of their study time.

Reply 1: Thank you for the kind words and suggestion. We have added further elaboration on the potential influence of the COVID-19 pandemic on our results, “On the whole, upon examining the clustering of tweets, it is evident that the dominant topic identified in our analysis centers around “campaigns”. The prominence of this topic could be contributed in part by the COVID-19 pandemic, which spanned a significant portion of the study’s duration. Amidst the pandemic, there was a widespread push for promoting the uptake of COVID-19 vaccines, followed closely by advocacy for the seasonal flu shots in many countries. For the 2021-2022 influenza season, the World Health Organisation (WHO) and numerous European countries promoted the safe and efficacious coadministration of influenza and COVID-19 vaccines [39].”

Comment 2: Despite the novelty of its methodology, the results are quite predictable and I do not consider them to contribute novelties to the current state of the subject. Can as authors justify that practical contribution is derived from the results of their study?

Reply 2: Thank you for the comment. We have provided further elaboration on the practical contribution derived from the results of our study, “The study offers a structured and comprehensive analysis of how influenza vaccination is discussed on a major social media platform during a significant period in global health, providing valuable data for future health communication strategies. While the analysis did not reveal any significant clusters of tweets containing misinformation or misunderstandings, the prevailing public messaging exhibited a narrow scope. Future messaging should encompass broader aspects, including disseminating comprehensive knowledge about influenza and the wide-ranging advantages of vaccination (capability), advocating for diverse means of vaccine access (opportunity), and bolstering motivation through greater incentivisation and social support for vaccination (motivation).”

Comment 3: The exclusion of tweets written in Spanish excludes almost half of the world’s population and is an important limitation of their study.

Reply 3: Thank you for the comment. We have now added this in our discussion of study limitations, “Related to this, only English-language tweets were analyzed in this study because English is one of the most widely used languages on Twitter and focusing on a single language ensures consistency in language processing techniques and reduces the complexity introduced by multilingual data analysis. However, it is important to note that this choice may introduce a bias towards English-speaking regions and their specific cultural, political, and health contexts.”

Comment 4: Other limitations exist with this study. Human language is not always explicit, and tweets are limited.

Reply 4: Thank you for the comment. We agree with the reviewer and have added this in our discussion of study limitations, “Twitter’s imposed character limit (previously 140, now 280) restricts the amount of information that can be shared in a single tweet. In terms of public messaging, this has inherent constraints, and in our analyses, can lead to oversimplification or loss of context, making it challenging to capture and convey certain nuanced information regarding influenza vaccines and vaccine hesitancy. Human language is not always explicit and tweets are limited.”

Comment 5: In addition, these data were collected using keyword searches and may have missed participants.

Reply 5: Thank you for the comment. We agree with the reviewer and have added this in our discussion of study limitations, “Second, the tweets used in the present dataset were filtered based on specific keyword searches, potentially leading to some selection biases. The search criteria might predominantly spotlight tweets from mainstream organisations endorsing vaccination. As an illustration, tweets employing hashtags like #BigPharma and #StopMandatoryVaccines could disseminate misleading information or demonstrate 'anti-vax' stances on flu shots without directly mentioning “flu” or “influenza”. Such tweets might be overlooked in our search parameters. This could mean our analysis might not capture these nuanced discussions and alternate messages.”

Comment 6: From the ethical point of view it is understood that all users of social networks authorize their content to be analyzed, however there are many vaccine users who do not use social networks to comment on them. There is significant selection bias.

Reply 6: Thank you for the comment. We agree with the reviewer and have added this in our discussion of study limitations, “This potential sampling bias may restrict the applicability of our findings to more diverse populations. Moreover, there are also vaccine users who do not use social media.”

Round 2

Reviewer 3 Report

Thank you for your answers and to try to improve the new version of the manuscript. There are methodological problems that they cannot solve and therefore they mention it as limitations of the study.

Author Response

Thank you for your kind words. Yes, we have acknowledged the methodological limitations of our study.